# Structural insights into tRNA recognition of the human FTSJ1-THADA complex
Kensuke Ishiguro [1,2], Atsushi Fujimura [3,4] & Mikako Shirouzu [1] ✉

tRNA undergoes various post-transcriptional modifications in the anticodon loop. FTSJ1, a protein conserved among most eukaryotes, mediates 2'-O-methylations at position 32 (Nm32) or position 34 (Nm34), complexed with THADA or WDR6, respectively. These methylations are crucial for accurate translation and cellular growth. FTSJ1 mutations are associated with non-syndromic X-linked intellectual disability. Although the structure of the FTSJ1-WDR6 complex in yeast has been solved, the structural details of the FTSJ1-THADA complex formation and substrate recognition remain unclear. Herein, using cryo-electron microscopy, we solve the high-resolution structure of FTSJ1-THADA with or without a tRNA substrate. FTSJ1 binds to THADA via its C-terminal region, with a unique interaction mode distinct from the FTSJ1-WDR6 complex. The tRNA substrate is anchored inside THADA, and key THADA residues for THADA-tRNA interaction are identified via structural and biochemical analyses. These findings demonstrate how FTSJ1 and THADA form a complex to mediate Nm32 modification in various tRNAs.

Approximately 150 RNA modification species have been identified in various RNA molecules across all domains of life[1]. Among RNA species, tRNA contains the widest variety of post-transcriptional modifications at various positions, especially in the anticodon loop[1,2]. Although tRNA modifications in the core region are involved in tRNA stability and metabolism[3,4], modifications in the anticodon loop play a critical role in translational fidelity and efficiency[2,5]. The most diverse modifications are found at the first position of the anticodon (position 34, known as the wobble position) and at the 3'-adjacent position of the anticodon (position 37)[2,5,6]. The molecular details and physiological functions of these modifications have been extensively studied[7–13]. Other than these two positions, 2'-O-methylations (Nm) at position 32 are widely found across all domains of life. In bacteria and archaea, Nm32 is introduced by the SPOUT family methyltransferase TrmJ[14]. In eukaryotes, Nm32 is introduced by the class I-type Rossmann fold methyltransferase FTSJ1 (also called Trm7 in yeast)[15].

FTSJ1 is a bifunctional protein that forms a complex with THADA (also called Trm732 in yeast) or WDR6 (also called Trm734 in yeast) to mediate Nm32 or Nm34 formation, respectively[16]. These three proteins are widely conserved in eukaryotes, except in several species belonging to the Stramenopiles–Alveolata–Rhizaria (SAR) supergroup or cryptophytes (Supplementary Fig. 1). Loss of these two modifications results in growth retardation in yeast[16,17] or human cell cultures[18], and reduced body weight in mice[19,20] or *Drosophila melanogaster*[21]. FTSJ1 is implicated in non-

syndromic X-linked intellectual disability (NSXLID)[22], and loss of FTSJ1 causes neurological abnormalities and abnormal behavior[19–21]. These reports highlight the physiological significance of Nm32 and Nm34 in almost all eukaryotes. Notably, *D. melanogaster* has two FTSJ1 homologs: one (FTSJ1_Nm32) modifies Nm32 and the other (FTSJ1_Nm34) modifies Nm34, in complexes with THADA or WDR6, respectively[21]. FTSJ1_Nm32 knockout (KO) impairs small RNA silencing, and reduces the mobility and lifespan[21].

Currently, the primary cause of these phenotypes is thought to be tRNA[Phe] dysfunction, as tRNA[Phe] overexpression substantially rescues the growth phenotype of FTSJ1 KO in yeast[16,17] or human cells[18]. tRNA[Phe] possesses both Nm32 and Nm34, prerequisites for adding wybutosine (yW) derivatives at position 37 both in yeast[16,17] and mammals[18,20,22] (Fig. 1a). Loss of these three modifications causes instability[18,20,21], charging defect[23], and lower decoding efficiency[18,20] of tRNA[Phe], resulting in global translational fluctuation[18,20]. In addition to tRNA[Phe], FTSJ1-THADA introduces Nm32 in approximately 10 tRNA species, whereas FTSJ1-WDR6 introduces Nm34 in several tRNA species in human[18,20,24,25] (Supplementary Table 1). However, the physiological roles of Nm32 and Nm34 in these tRNAs remain unclear.

Recently, the structure of the yeast FTSJ1-WDR6 complex was elucidated using X-ray crystal structure analysis[26]. S-adenosyl-L-methionine (SAM) binds to the active site of FTSJ1, whereas the C-terminal region of

[1]Laboratory for Protein Functional and Structural Biology, RIKEN Center for Biosystems Dynamics Research, Yokohama, Kanagawa, Japan. [2]Department of Chemistry and Biotechnology, Graduate School of Engineering, The University of Tokyo, Tokyo, Japan. [3]Department of Cellular Physiology, Okayama University Graduate School of Medicine, Dentistry, and Pharmaceutical Sciences, Okayama City, Okayama, Japan. [4]Neutron Therapy Research Center, Okayama University, Okayama City, Okayama, Japan. ✉e-mail: mikako.shirouzu@riken.jp

FTSJ1 (Asn233-Ser259, Ser217-Asp240 in human numbering) is crucial for binding to WDR6[26]. A similar region in human FTSJ1 (Phe222-Ser243) is essential for binding to WDR6[18], suggesting that the binding mechanism between FTSJ1 and WDR6 is conserved among eukaryotes. Although the structure of the complex with tRNA has not yet been determined, biochemical analysis suggests that yeast FTSJ1-WDR6 recognizes Nm32, 1-methylguanosine ($m^1G$) 37 (an intermediate of yW derivatives), and the D-loop of tRNA to efficiently introduce Nm34[18,26].

However, the molecular mechanism by which FTSJ1-THADA introduces the Nm32 modification remains unclear. Despite the low sequence similarity among THADA homologs, human THADA can substitute for yeast Trm732 in vivo[17], suggesting a conserved binding mechanism between FTSJ1 and THADA in eukaryotes. A recent study examined several conserved THADA motifs (Supplementary Fig. 2b) for their physiological function[27]. Although the importance of conserved motif 3 (Asn1401-Leu1408) is not well established, conserved motifs 1 (Leu1100-Gly1107) and 2 (Ala1157-Pro1165) are essential for the physiological functions of THADA. However, the structure of the FTSJ1-THADA complex has not been elucidated.

Thus, in this study, we aimed to solve the high-resolution structure of the FTSJ1-THADA complex, with and without a tRNA substrate.

## Results

### Cryo-electron microscopy analysis of FTSJ1-THADA

Human FTSJ1 and THADA were co-expressed in Freestyle HEK293F cells and purified using anti-FLAG beads and size-exclusion chromatography. Both FTSJ1 and THADA were detected in the fractions corresponding to a molecular weight of approximately 260 kDa in size-exclusion chromatography (Fig. 1b), demonstrating the stable formation of the αβ heterodimer.

Purified FTSJ1-THADA was subjected to cryo-electron microscopy (cryo-EM) analysis (Supplementary Fig. 3a). In Micrographs, a ring-like structure with a diameter of approximately 100 Å was observed (Fig. 2a). Following particle picking, 2D classification, and several 3D classifications, a consensus map was obtained that aligned with the AlphaFold prediction of THADA[28,29] (Fig. 2b). THADA forms a hollow cylindrical structure with a flexible N-terminal tail-like domain (1–344), which showed weak density, suggesting that this region has high flexibility (Fig. 2b). On the top of THADA, an unassigned weak density matching the size and shape of FTSJ1 was observed (Fig. 2c). To visualize the N-terminal tail-like domains of THADA and FTSJ1, multiple 3D classifications were performed, focusing on the FTSJ1 moiety (Supplementary Fig. 3a). Finally, five cryo-EM maps with relatively high FTSJ1 density were obtained at 3.3–4.1 Å resolution (Supplementary Fig. 3a).

The arrangement of N-terminal tail-like domains differed greatly among the structures (Fig. 2d–f, Supplementary Fig. 5). In class 1, the N-terminal tail-like domain is located close to the cylindrical structure of THADA, limiting access to the interior (Fig. 2d), thus, we refer to this structure as the "closed form." In contrast, the N-terminal tail-like domain in class 2a–2d has notably shifted outward by 24–52 Å, measured from helix α1 at the end of the N-terminal tail-like domain (Fig. 2e, f, Supplementary Fig. 5). This outward shift makes the entrance of THADA completely accessible; therefore, these structures are called the 'open form.' We performed a 3D variability analysis using all THADA-like particles (Supplementary Movie 1), Class 1 (Supplementary Movie 2), and Class 2 particles (Supplementary Movie 3), to evaluate the conformational heterogeneity of these structures. This analysis revealed that the N-terminal tail-like domain of THADA was highly flexible, showing movement between two different conformations corresponding to the open and closed forms. Although we attempted to fit the alpha fold prediction of THADA and FTSJ1 to these maps, the maps did not achieve atomic resolution in the N-terminal tail-like domain and FTSJ1, owing to their high flexibility.

### Cryo-electron microscopy structure of FTSJ1-THADA with tRNA substrate

We then attempted to obtain the cryo-EM structure of FTSJ1-THADA with a tRNA substrate. Yeast FTSJ1-THADA introduced Cm32 into unmodified

tRNA$^{Phe}$ transcripts[15,26], whereas accurate methylation efficiency was not measured. Therefore, we anticipated that FTSJ1-THADA would bind to unmodified tRNA substrates. First, we incubated FTSJ1-THADA with human tRNA$^{Phe}$ transcripts and conducted cryo-EM analysis; however, tRNA-bound particles were not detected. Thereafter, we incubated FTSJ1-THADA with human tRNA$^{Phe}$ transcripts and S-adenosyl homocysteine (SAH), followed by cryo-EM analysis (Supplementary Fig. 3b). After the first classification, we identified a class containing a tRNA-like helix moiety within FTSJ1-THADA. This class underwent further image processing and model building, resulting in the reconstructed cryo-EM map of FTSJ1-THADA with substrate tRNA$^{Phe}$ and SAH at 2.66 Å resolution (PDB:8Y2O) (Fig. 3a, b, Supplementary Fig. 3b). Although most regions of FTSJ1-THADA and tRNA were successfully modeled, the C-terminal region of FTSJ1 (Asp208-Gly227, Asp253-Pro329), the central region of the DUF2428 domain of THADA (Glu973-Cys1042), and the 3′-terminal of substrate tRNA$^{Phe}$ (A73–A76) were not visible and could not be modeled (Fig. 3c). THADA forms a hollow cylindrical structure with FTSJ1 bound to the top, which completely fits the closed form of FTSJ1-THADA. Notably, the substrate tRNA$^{Phe}$ was located inside THADA, anchoring the anticodon loop to the methyltransferase center of FTSJ1 (Fig. 3d). The interior of THADA and the tRNA binding site of FTSJ1 are strongly positively charged (Fig. 3f, g), facilitating the binding of the substrate tRNA$^{Phe}$. The tRNA-bound FTSJ1-THADA was exclusively observed in the closed form (Supplementary Fig. 3b and Supplementary Movie 4), suggesting that THADA was stabilized in this state after tRNA binding. Notably, whereas the N-terminal tail-like domain of THADA contacts with a D-loop of tRNA (Fig. 3b) via the α10 helix, this contact is not allowed in the open forms, as indicated by superimposed models (Supplementary Fig. 6).

### Methyltransferase center of FTSJ1-THADA is located near C32 of the tRNA substrate

Thereafter, attention was directed to the methyltransferase center of FTSJ1 (Fig. 3d). SAH bound to the active site of FTSJ1 via hydrogen bonds with four residues (Ser25, Ser54, Asp75, and Asp91) and several hydrophobic interactions (Fig. 3d, Supplementary Figs. 2a, 7a). This binding mode was completely conserved in yeast FTSJ1-WDR6 (Fig. 3e)[26], indicating that the methylation mechanism of FTSJ1 is consistent, whether bound to THADA or WDR6. Moreover, the 2′-OH group of C32, the methylation target of FTSJ1-THADA, is located 3.3 Å from the sulfur atom of SAH (Fig. 3d). Four catalytic residues that are conserved in FTSJ1 families[30] (Lys28, Asp116, Lys156, and Glu191) surround C32 of the substrates tRNA$^{Phe}$ and SAH, with Lys156 forming a hydrogen bond with the 2′-OH group of C32, suggesting its role as a catalytic center for the methyl transfer reaction.

### FTSJ1-THADA complex formation

Thereafter, we examined the interaction sites between FTSJ1 and THADA (Fig. 4a). Notably, the N-terminal region of FTSJ1 had a completely different conformation from that of the previously solved structure of the yeast FTSJ1-WDR6 complex (Fig. 4a, b, Supplementary Fig. 2a)[26]. The first half of the WDR6 binding region (Ser217-Asp240 in human numbering) of FTSJ1 was nearly invisible in the FTSJ1-THADA complex, and the α7 helix of FTSJ1 was composed of different residues between the two structures (Fig. 4a, b). The THADA-binding site of FTSJ1 (Cys238-Tyr250) interacted with α62 helix and the loop between α62 and α63 of THADA (Supplementary Fig. 7), forming several hydrogen bonds, including Cys238-Pro1358, Gly239-Arg1363, Asp240-Asn1401, Ser249-Cys1393, Tyr250-Gln1400, and Tyr250-Gln1449, alongside a π–π stacking interaction of Tyr250-Phe1398 (Fig. 4c, d). Notably, the conserved motif 3 (Asn1401-Leu1408) of THADA participates in binding to FTSJ1, although its physiological importance has not been well established[27]. Additionally, Asp126 of FTSJ1 formed a hydrogen bond with Arg1368 of THADA, and Tyr130 of FTSJ1 formed a π–π stacking interaction with Phe1321 of THADA (Fig. 4e).

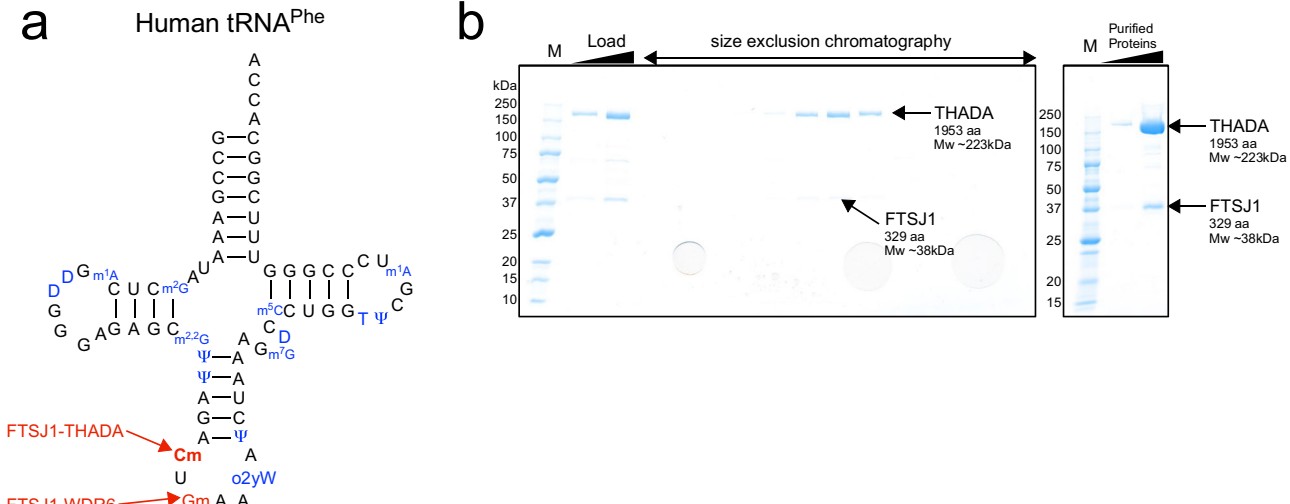

**Fig. 1 | Purification of recombinant human FTSJ1-THADA. a** Cloverleaf structure of human tRNA<sup>Phe</sup> with tRNA modifications. 2′-*O*-methyl-cytidine (Cm) at 32 and 2′-*O*-methyl-guanosine (Gm) at 34 are indicated in red. 2-methylguanosine (m²G) at 10, 1-methyladenosine (m¹A) at 14 and 58, dihydrouridine (D) at 16,17 and 47, 2,2-dimethylguanosine (m²,²G) at 26, pseudouridine (Ψ) at 27, 28, 39 and 55, peroxywybutosine (o2yW) at 37, 7-methylguanosine (m⁷G) at 46, 5-methylcytidine (m⁵C) at 49, and 5-methyluridine (T) at 54 are indicated in blue. 2′-*O*-methylation at 32 and 34 is mediated by FTSJ1-THADA and FTSJ1-WDR6, respectively. **b** SDS-PAGE of FTSJ1-THADA during size-exclusion chromatography (left) and purified FTSJ1-THADA (right). The gel was stained with Coomassie Brilliant Blue. An unedited gel image is also provided as Supplementary Fig. 11.

**Fig. 2 | Cryo-EM structure of FTSJ1-THADA.**
**a** Micrograph image of FTSJ1-THADA. White arrows indicate particles. **b**, **c** Overall view (**b**) and close-up view at the top (**c**) of the consensus map of FTSJ1-THADA superimposed with the AlphaFold model of THADA[28,29]. THADA is shown in green, and the N-terminal tail-like domain of THADA (1–344) is highlighted in light green. **d**, **e** Each map is superimposed on the closed-form model of FTSJ1-THADA. The bottom images show a close-up view of the top of THADA. FTSJ1 is indicated in orange, THADA in green, the N-terminal tail-like domain of THADA (1–344) in light green, and the α1 helix of THADA (18–25) in red. **f** Comparing the position of the α1 helix of THADA (18–25) among five structures. Distances between His18 in the α1 helix of class 1 and the corresponding position in the other structures are displayed.

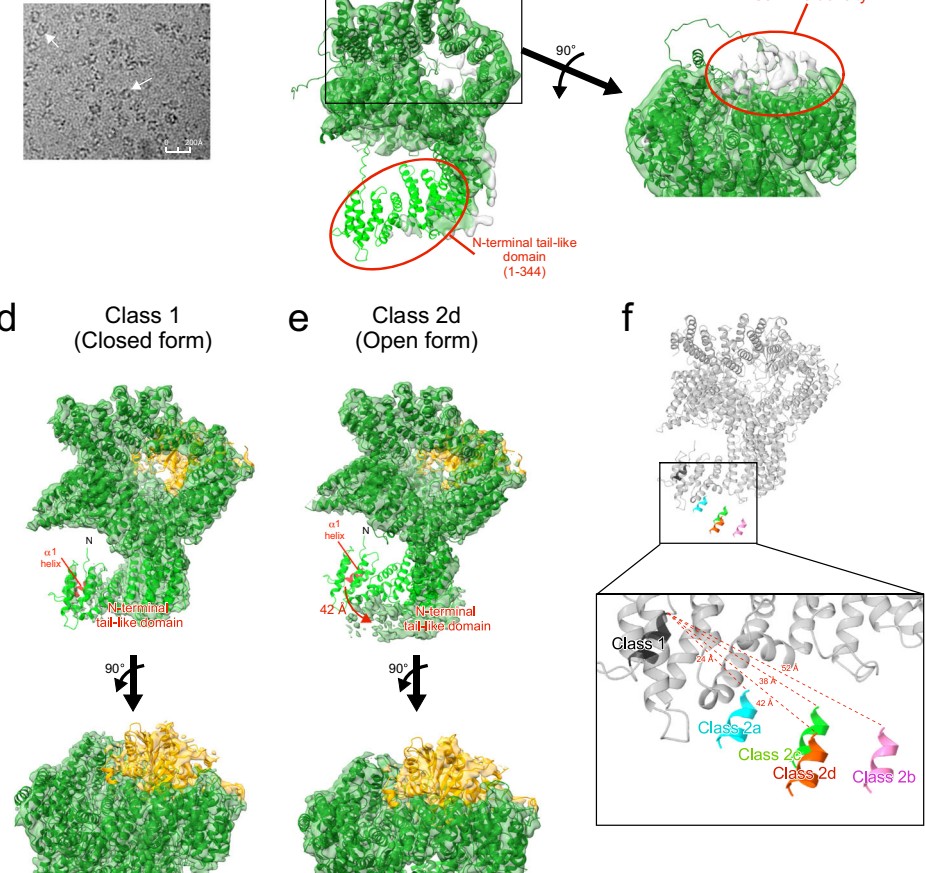

**Fig. 3 | Cryo-EM structure of FTSJ1-THADA with tRNA substrate. a, b** Overall structure of FTSJ1-THADA with substrates tRNA^Phe and *S*-adenosyl homocysteine (SAH) (PDB:8Y2O, EMD-38859). FTSJ1, THADA, and tRNA are shown in orange, green, and cyan, respectively. **a** The map is superimposed on the model. **b** The model alone. The right image is rotated 90° along the horizontal axis. The N-terminal tail-like domain of THADA (1–344) is highlighted in light green. SAH and C32 are highlighted as blue and red spherical models, respectively. Key residues required for tRNA-THADA interaction are highlighted as purple (tRNA residues) and dark orange (THADA residues) spherical models, respectively. **c** Schematic representations of FTSJ1 and THADA. FTSJ1 and THADA are shown in orange and green, respectively. Conserved domains are depicted as boxes. Regions or residues required for binding to FTSJ1, THADA, tRNA, or SAH are colored orange, green, cyan, or blue, respectively. Catalytic residues conserved in the FTSJ1 families[30] are highlighted. Residues subjected to mutation studies are shown in bold. The invisible regions in the map are depicted as gray boxes or dotted lines. **d** Methyltransferase core of FTSJ1 in the FTSJ1-THADA-tRNA complex (PDB:8Y2O). The amino acid residues responsible for interactions with SAH are represented as stick models. The hydrogen bonds with SAH are indicated by dotted lines. **e** Methyltransferase core of FTSJ1 in the yeast FTSJ1-WDR6 complex (PDB:6JPL)[26]. The amino acid residues responsible for interactions with *S*-adenosyl methionine (SAM) are represented as stick models. Numbers in parentheses indicate residue numbers in human FTSJ1. The hydrogen bonds with SAM are indicated by dotted lines. **f, g** Electrostatic surface potential of FTSJ1-THADA generated by the Adaptive Poisson–Boltzmann Solver (APBS) software[69] overlaid on the human tRNA^Phe (cyan) and SAH (gray) (PDB:8Y2O). Key tRNA residues required for tRNA-THADA interaction and C32 are highlighted as magenta and orange spherical models, respectively. The positively and negatively charged areas are colored blue and red, respectively. **f** Overall view of FTSJ1-THADA from the side (left) or below (right). **g** Close-up view of FTSJ1 with a tRNA substrate.

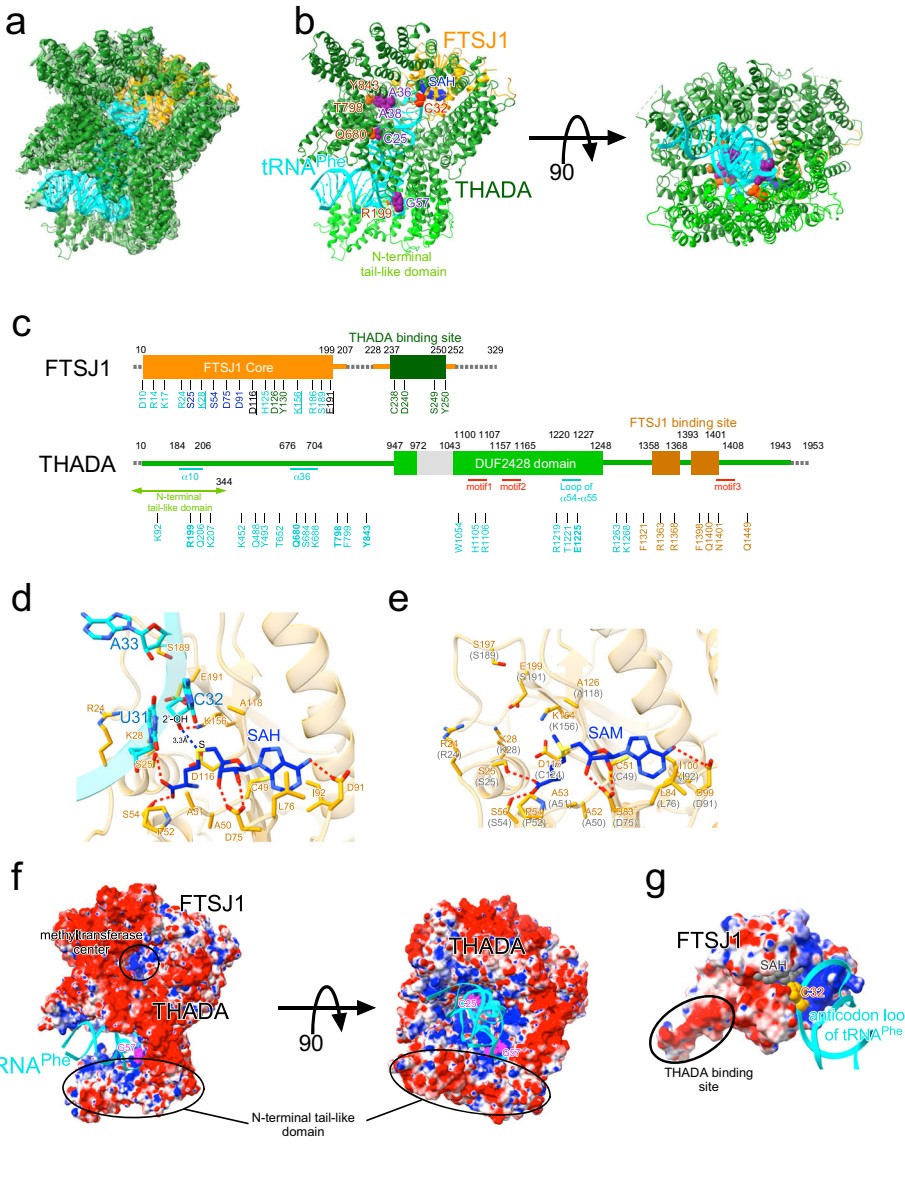

## tRNA recognition by FTSJ1-THADA

Finally, we investigate the conformation of tRNA and its interactions with FTSJ1-THADA. Compared to the structure of tRNA located at the P-site of the ribosome[7], the tRNA bound to FTSJ1-THADA exhibited a shift in the anticodon loop of approximately 10 Å (Fig. 5a). This shift, resulting from the interactions between FTSJ1-THADA and the substrate tRNA^Phe (summarized in Fig. 5b), places C32 of the tRNA near the methyltransferase center of FTSJ1 and SAH. FTSJ1 primarily recognizes the backbone of the anticodon loop of the tRNA substrate (Fig. 5b, c). These interacting residues, primarily located in the methyltransferase center of FTSJ1, were also present in the α1 helix (Supplementary Fig. 2a, Supplementary Fig. 7a). In contrast, THADA recognized the anticodon loop, tRNA core, and elbow regions (Fig. 5b–d). tRNA-recognizing residues were observed primarily in the cylindrical core (α36 helix, conserved motif 1 (Leu1100-Gly1107), loop between α54 and α55), and also in N-terminal tail-like domain (α10 helix) (Supplementary Fig. 2b, Supplementary Fig. 7b). Numerous THADA residues recognized the backbone of the tRNA substrate; however, several residues, both in THADA cylindrical core (Gln680 with C25, Thr798 with A36 and A38, Arg 1105 and His1106 with A35, and Glu1225 with G20) and

N-terminal tail-like domain (Arg199 with G57), forming hydrogen bonds with the bases of the tRNA substrate (Fig. 5b–d). Several residues in the THADA cylindrical core (Phe799 with A38, Tyr843 with A36, and Trp1054 with A35) engage in π–π stacking interactions with the tRNA bases, indicating their importance for tRNA recognition and selection by FTSJ1-THADA. Consistent with this notion, His1105 is located in conserved motif 1 (Leu1100-Gly1107), crucial for the physiological function of THADA[27]. To elucidate the importance of the other residues, we evaluated the methyltransferase activity of the recombinant FTSJ1-THADA mutants by measuring their SAM consumption using total RNAs purified from FTSJ1 KO cells as the substrate (Fig. 5e). In this assay, SAH produced by the methyltransferase activity of FTSJ1 was degraded to adenosine and homocysteine by adenosylhomocysteinase (AHCY). By detecting the amount of homocysteine with the fluorescence intensity of the thiol fluorescent probe IV, the methyltransferase activity of FTSJ1-THADA was semiquantitatively quantified. The fluorescence intensity was observed when wildtype FTSJ1-THADA was added (lane 1), but not when either one was missing (lanes 2 and 3), confirming the validity of this experimental system (Fig. 5e). Notably, Glu1225 mutation caused a modest decrease in the

**Fig. 4 | Structural insights into FTSJ1-THADA complex formation. a** Binding regions between FTSJ1 and THADA in the FTSJ1-THADA-tRNA complex (PDB:8Y2O). The THADA-binding site of FTSJ1 is indicated in red, the FTSJ1-binding site of THADA is in green, and the conserved motif 3 (Asn1401-Leu1408) of THADA[27] is in light blue. Amino acid residues responsible for the interactions between FTSJ1 and THADA are represented as stick models. **b** Binding regions between yeast FTSJ1 and WDR6 (PDB:6JPL)[26]. The WDR6 binding site of FTSJ1 is shown in red. Numbers in parentheses indicate residue numbers in human FTSJ1. **c–e** Close-up view of the interactions between FTSJ1-THADA (PDB:8Y2O). Hydrogen bonds are depicted as red dotted lines, and π–π stacking interactions are depicted as blue dotted lines.

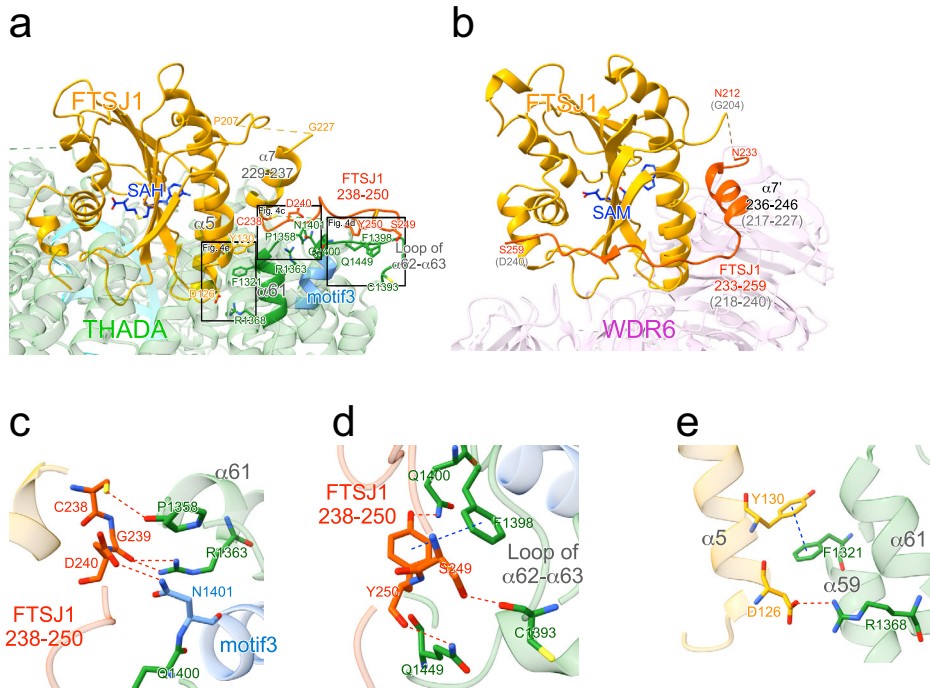

methyltransferase activity of FTSJ1-THADA (lane 8), whereas other mutations in Gln680, Thr798, and Tyr843 in THADA cylindrical core and Arg199 in N-terminal tail-like domain almost completely abolished methyltransferase activity (lane 4-7). We confirmed these mutants did not affect the interaction with FTSJ1 by immunoprecipitation experiments (Supplementary Fig. 8), implying that these residues are crucial for tRNA recognition by FTSJ1-THADA. Although conserved motif 2 (Ala1157-Pro1165) is crucial for the physiological function of THADA[27], it did not directly interact with FTSJ1 or the tRNA substrate; instead, it was located near the methyltransferase center of FTSJ1 and the anticodon loop of tRNA (Fig. 5f). This proximity implied that motif 2 is also important for tRNA binding; however, its precise role remains unclear.

## Discussion

In this study, we performed cryo-EM analysis of FTSJ1-THADA with and without a tRNA substrate, identifying the critical residues required for tRNA recognition by FTSJ1-THADA. Based on these findings, we hypothesized that tRNA methylation is mediated by FTSJ1-THADA (Fig. 6). When the tRNA substrate was unbound, the N-terminal tail-like domain was not fixed and moved freely between open and closed forms, whereas the tRNA substrate could bind to FTSJ1-THADA only in the open form. Upon entering FTSJ1-THADA, the anticodon loop of the tRNA substrate is bent and anchored around the FTSJ1 methyltransferase via various electrostatic and hydrophobic interactions (Fig. 6). Furthermore, after the tRNA substrate was anchored, the N-terminal tail-like domain of THADA was stabilized in the closed form and the interaction with tRNA substrate, such as Arg199 with G57, promoted the methyltransferase reaction by FTSJ1-THADA. The absence of SAH resulted in unstable tRNA binding to FTSJ1-THADA, suggesting that SAM binding was a prerequisite for stable binding with the tRNA substrate. In the future, kinetic analysis, such as by an electrophoretic mobility shift assay, can clarify the detailed reaction mechanism of FTSJ1-THADA.

In contrast to FTSJ1-WDR6, which required m[1]G37 and Cm32 modifications for stable binding to tRNA[Phe 18,26], FTSJ1-THADA bound stably to the tRNA substrate without any modifications. This implies that the introduction of Nm32 by FTSJ1-THADA occurred before the introduction of Nm34 by FTSJ1-WDR6 during tRNA maturation. To evaluate the order of introduction of these methylations, it is necessary to examine the tRNA modification status in the tRNA precursor. Whether the presence of Nm34 or m[1]G37 inhibits tRNA recognition by FTSJ1-THADA merits further investigation.

Notably, the binding mode between FTSJ1 and THADA differs from that of FTSJ1-WDR6[26]. This indicated that the C-terminal region of FTSJ1 can adopt two different conformations in response to two adapter proteins, WDR6 and THADA. Unlike other organisms, *D. melanogaster* has two FTSJ1 homologs corresponding to WDR6 or THADA binding, respectively[21]. As stated[21], the sequences of the N-terminal methyltransferase core of these two homologs are almost identical, but the sequences of their C-terminal regions differ (Supplementary Fig. 2a). In particular, among the FTSJ1 residues participating in interaction with THADA (Fig. 4), Cys238 is not conserved in FTSJ1_Nm34 of *D. melanogaster*, suggesting that this residue is crucial for binding with THADA, and not with WDR6.

In humans, FTSJ1-THADA mediates Cm32 formation on various tRNA substrates. We extracted the sequences of Nm32-containing tRNAs reported in two studies[18,25] and compared their sequence bias with that of all human tRNAs (Supplementary Data 1). Among the tRNA positions recognized by FTSJ1-THADA, purines at positions 36 and 38 were strongly biased toward FTSJ1-THADA substrates (Supplementary Fig. 9). A36 and A38 of tRNA formed multiple hydrogen bonds and π–π stacking interactions with surrounding THADA residues such as Tyr843 and Thr798 (Fig. 5c), which would be lost if these positions were substituted with pyrimidine residues. Moreover, mutations in Tyr843 or Thr798 of THADA significantly decreased the methyltransferase activity of FTSJ1-THADA (Fig. 5e), supporting the idea that purines at positions 36 and 38 are positive determinants of tRNA recognition by FTSJ1-THADA. The purine at position 20 was also marginally biased in FTSJ1-THADA substrates (Fig. S8), whereas loss of the G20-Glu1225 interaction (Fig. 5d) by the Glu1225Ala mutant had a modest effect on FTSJ1-THADA activity (Fig. 5e). This suggests that the Glu1225-G20 interaction is not critical determinant for tRNA binding, although it is a positive factor. The interaction between U16 and Lys92, which is also present in the D-loop, may compensate for the loss of the Glu1225-G20 interaction.

Notably, THADA has a high degree of heterogeneity among its homologs, with a 3-fold difference in length observed (Supplementary

**Fig. 5 | Structural insights into tRNA recognition by FTSJ1-THADA. a** tRNA[Phe] and SAH in FTSJ1-THADA (PDB:8Y2O) (cyan) is superimposed with tRNA[iMet] at the P site of the 70S ribosome (PDB:7Y7F) (gray). **b** tRNA recognition residues of FTSJ1 (orange) and THADA (green) are depicted in the secondary structure of human tRNA[Phe]. Highly conserved residues are underscored. Residues subjected to mutation studies are shown in bold. Hydrogen bonds with tRNA bases are depicted as red solid lines, hydrogen bonds with tRNA main-chains are depicted as red dotted lines, π–π stacking interactions with tRNA bases are depicted as blue solid lines. **c, d** tRNA recognition residues of FTSJ1 (orange) and THADA (green) around the anticodon loop (**c**) or other regions (**d**) of tRNA[Phe] (cyan). Conserved motif 1 (Leu1100-Gly1107) of THADA[27] is shown in red. Residues subjected to mutation studies are shown in bold. The amino acid residues responsible for the interactions with tRNA are represented as stick models. The right panels show a close-up view of the interactions with tRNA. Hydrogen bonds are depicted as red dotted lines, and π stacking interactions are depicted as blue dotted lines. **e** The methyltransferase activity of recombinant FTSJ1-THADA mutants in vitro. Data are presented as the mean ± standard deviation ($n = 3$, technical replicates). *$P < 0.05$ (One-way ANOVA with Tukey's multiple comparison test). The raw data of this assay is also provided as Supplementary Data 2. **f** Close-up view of the conserved motif 2 (Ala1157-Pro1165) of THADA (magenta).

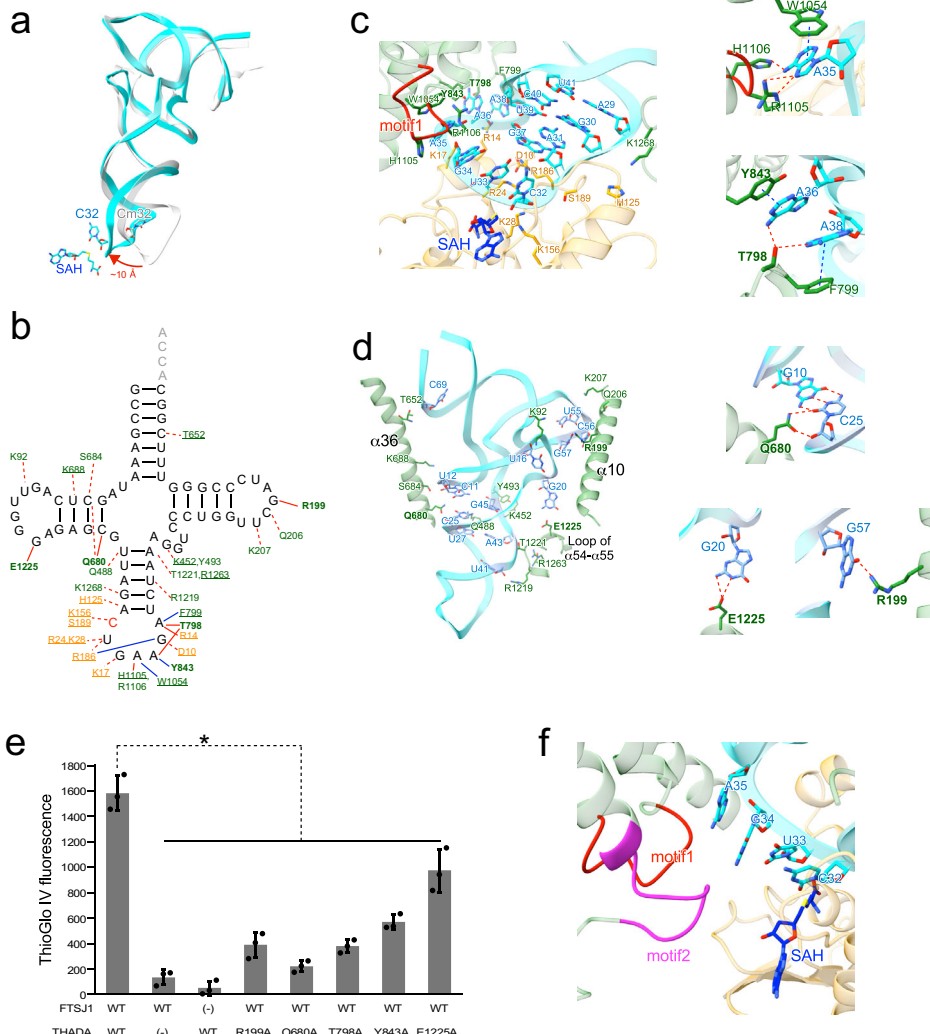

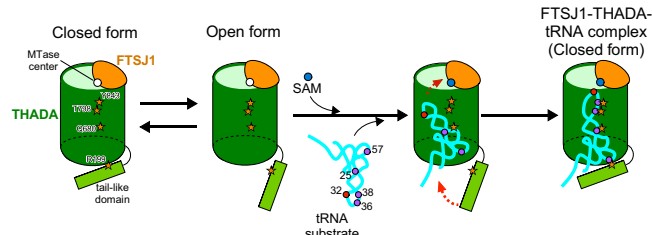

**Fig. 6 | Hypothetical structural mechanism of tRNA recognition by FTSJ1-THADA.** The hollow cylindrical core of THADA, the N-terminal tail-like domain of THADA, FTSJ1, and tRNA are indicated in green, light green, orange, and cyan, respectively. Key residues required for tRNA-THADA interaction are highlighted as a purple circle (tRNA residues) and a dark orange star (THADA residues), respectively. SAM and C32 are highlighted as blue and red circles, respectively.

Fig. 1). This variability raises questions about the conservation of THADA's function across different homologs. To confirm this, we obtained AlphaFold models of shorter THADA homologs (*Saccharomyces cerevisiae* and *Caenorhabditis elegans*) and superimposed them onto our human FTSJ1-THADA model (Supplementary Fig. 10). Both these homologs exhibited a hollow cylindrical structure with a strongly positively charged interior, similar to that of human THADA, suggesting that they shared tRNA recognition mechanism (Supplementary Fig. 10b). However, they lacked an N-terminal tail-like domain present in human THADA (Supplementary

Fig. 10a), and the critical residues for tRNA binding in human THADA are not highly conserved among these homologs (Fig. 5b, Supplementary Fig. 2b). This suggested that the details of the tRNA recognition mechanism may differ among THADA homologs, potentially accounting for the discrepancy in the number of substrate tRNAs between yeast (three species)[16] and human THADA (approximately 10 species) (Supplementary Table 1).

Loss of Nm32 and/or Nm34 by FTSJ1 mutation is implicated in non-syndromic X-linked intellectual disability (NSXLID)[19–22]. Notably, a novel missense allele of FTSJ1 (A26P) associated with intellectual disability reduces the frequency of Nm32, but not Nm34[22]. A26 of FTSJ1 is located close to the tRNA substrate and SAM, but the SAM recognition mechanism is conserved between FTSJ1-THADA and FTSJ1-WDR6 (Fig. 3d, e). Therefore, the difference in the tRNA recognition mechanism between FTSJ1-THADA and FTSJ1-WDR6 may cause the reduction in the frequency of Nm32 by A26P. Elucidation of the FTSJ1-WDR6-tRNA structure can enhance the understanding of this aspect.

Although THADA knockdown does not cause any growth defects in yeast[16,17], THADA is associated with various cellular processes and diseases in higher eukaryotes. THADA was originally identified as a target for chromosomal rearrangements in thyroid adenomas[31]. The rearrangement could produce a chimeric THADA mRNA that lacks FTSJ1-binding regions. However, a subsequent study suggested that the primary driver of cancer cell malignancy is the upregulation of downstream genes rather than the production of this chimeric protein[32]. Single-nucleotide polymorphisms (SNPs) related to various cancers have been detected in the THADA gene[33–35]. Notably, THADA overexpression increases cancer

**Table 1 | Cryo-EM data collection, refinement, and validation statistics**

| | FTSJ1-THADA with substrate tRNA (EMD-38859) (PDB 8Y2O) | FTSJ1-THADA (Class 1) (EMD-61701) | FTSJ1-THADA (Class 2a) (EMD-61702) | FTSJ1-THADA (Class 2b) (EMD-61703) | FTSJ1-THADA (Class 2c) (EMD-61704) | FTSJ1-THADA (Class 2d) (EMD-61705) |
|---|---|---|---|---|---|---|
| Data collection and processing | | | | | | |
| Magnification | 105,000 | 105,000 | 105,000 | 105,000 | 105,000 | 105,000 |
| Voltage (kV) | 300 | 300 | 300 | 300 | 300 | 300 |
| Electron exposure (e–/Å²) | 50 | 50 | 50 | 50 | 50 | 50 |
| Defocus range (μm) | 0.5–2.5 | 0.5–2.5 | 0.5–2.5 | 0.5–2.5 | 0.5–2.5 | 0.5–2.5 |
| Pixel size (Å) | 0.83 | 0.8285 | 0.8285 | 0.8285 | 0.8285 | 0.8285 |
| Symmetry imposed | C1 | C1 | C1 | C1 | C1 | C1 |
| Initial particle images (no.) | 7,275,741 | 5,314,179 | 5,314,179 | 5,314,179 | 5,314,179 | 5,314,179 |
| Final particle images (no.) | 1,854,516 | 281,340 | 126,372 | 83,420 | 107,538 | 203,303 |
| Map resolution (Å) | 2.66 | 3.31 | 3.85 | 4.06 | 3.78 | 3.71 |
| FSC threshold | 0.143 | 0.143 | 0.143 | 0.143 | 0.143 | 0.143 |
| Refinement | | | | | | |
| Initial model used (PDB code) | FTSJ1(Q9UET6) and THADA (Q6YHU6) in Alphafold Database tRNA in 7y7f | | | | | |
| Model resolution (Å) | 2.7 | | | | | |
| FSC threshold | 0.5 | | | | | |
| Map sharpening B factor (Å²) | −50 | −50 | −50 | −50 | −50 | −50 |
| Model composition | | | | | | |
| Non-hydrogen atoms | 17385 | | | | | |
| Protein residues | 2011 | | | | | |
| RNA residues | 72 | | | | | |
| Ligands | 72 (Mg), 1 (Zn), 1(SAH) | | | | | |
| B factors(mean) (Å²) | | | | | | |
| Protein | 68.16 | | | | | |
| RNA | 62.58 | | | | | |
| Ligand | 67.35 | | | | | |
| R.m.s. deviations | | | | | | |
| Bond lengths (Å) | 0.009 | | | | | |
| Bond angles (°) | 0.743 | | | | | |
| Validation | | | | | | |
| MolProbity score | 1.79 | | | | | |
| Clashscore | 7.82 | | | | | |
| Poor rotamers (%) | 0.06 | | | | | |
| Ramachandran plot | | | | | | |
| Favored (%) | 94.71 | | | | | |
| Allowed (%) | 5.24 | | | | | |
| Disallowed (%) | 0.05 | | | | | |

growth and malignancy via augmenting the mechanistic target of rapamycin (mTOR) pathway or upregulating PD-L1[36–38]. Furthermore, the knockdown of THADA represses cancer growth[38]. Consistent with this finding, FTSJ1 has also been implicated in cancer, albeit with conflicting results[39–41].

THADA has a strong relationship with type 2 diabetes (T2DM). SNPs related to T2DM (Thr1187Ala) have been reported[42,43], and THADA expression is strongly affected by hyperglycemia[44–46]. Chronic over-expression of THADA reduces insulin secretion by inhibiting the endoplasmic reticulum Ca²⁺ pump (SERCA) and inducing apoptosis in pancreatic β-cells, leading to T2DM symptoms[45,47]. Additionally, THADA has been identified as a hotspot for SNPs related to acclimatization[48,49], which implies THADA is involved in heat production. Consistent with this, THADA knockdown results in a cold-sensitive growth phenotype in both plants[50] and animals[51], attributed to dysregulation of the mTOR pathway and SERCA, respectively. Notably, THADA directly interacts with SERCA[45,51]; however, its complex has not yet been elucidated. The SNP site associated with T2DM (Thr1187)[42] is not located within the binding site for tRNA or FTSJ1, suggesting it may play an important role in binding to SERCA.

While SNPs related to polycystic ovary syndrome have been identified in the intron of *THADA*[33,52,53], the expression of THADA does not change during gestation[54], and THADA knockdown does not affect fertility in mice[55]. SNPs related to multiple sclerosis[56], non-syndromic cleft lip with or without cleft palate[57], and migraine[58] have also been detected near *THADA*, although their physiological implications remain unclear. Although THADA is reportedly involved in various cellular processes, as mentioned above, the existing studies do not analyze the Nm32 methylation frequencies in tRNAs. Consequently, whether these functions are derived from tRNA modification activity or other functions of THADA, such as interactions with SERCA, is unclear[45,51]. In this study, we elucidated the high-resolution structure of the tRNA-bound FTSJ1-THADA complex and the key interactions required for tRNA recognition by FTSJ1-THADA. These results allow us to speculate whether the pathogenetic mutations found in THADA or FTSJ1 are related to tRNA modification activity or other functions. Further physiological and biochemical analyses based on our structural analyses can elucidate the roles of FTSJ1-THADA and Nm32 in these diseases.

## Methods

### Purification of FTSJ1-THADA
The gene encoding FTSJ1 or THADA was PCR-amplified and cloned into a modified pcDNA3.4 vector (Thermo Fisher Scientific, MA, USA) using the primers listed in Supplementary Table 2. FLAG-tag and TEV protease cleavage sequences were inserted into the N-terminus of FTSJ1, and FLAG-tag, streptavidin-binding sequence, and TEV protease cleavage sequence were inserted into the N-terminus of THADA in the vector using the In-Fusion HD Cloning kit (Takara Bio, Shiga, Japan). These plasmids were co-transfected into FreeStyle HEK293F cells and cultured in FreeStyle 293 Expression Medium (Thermo Fisher Scientific, MA, USA) in a $CO_2$-controlled bioshaker $CO_2$-BR-180LF (TAITEC, Saitama, Japan) at 37 °C, 128 rpm, and a 20 mL/min 8% $CO_2$ supply. After 48 h of culture, 500 mL of the culture was harvested via centrifugation. Thereafter, the harvested cells were dissolved in 80 mL Lysis Buffer (20 mM Tris-HCl (pH 7.5), 150 mM NaCl, 2 mM dithiothreitol). Cells were sonicated with a Bioruptor BR-2 (Sonicbio, Kanagawa, Japan) at high power, 10 cycles of 60 s pulses each. The cell lysate was centrifuged at 15,000 rpm (20,400 × *g*) for 30 min at 4 °C, and the supernatant was collected and mixed with 6 mL anti-FLAG M2 agarose beads (50% v/v) (Sigma-Aldrich, MO, USA) using a rotator at 4 °C overnight.

The mixture was centrifuged at 1000 rpm (100 × *g*) for 1 min at 4 °C, and the supernatant was removed as a flow-through fraction. Subsequently, the beads were washed three times with 20 mL Lysis Buffer. Bound proteins were eluted using 15 mL Lysis Buffer containing 100 μg/mL 3× Flag peptide (Sigma-Aldrich, MO, USA). The protein was further purified using size-exclusion chromatography with a HiLoad 16/600 Superdex 200pg column (Cytiva, Tokyo, Japan) pre-equilibrated with Lysis Buffer. FTSJ1 and THADA were detected in the same fraction via sodium dodecyl sulfate–polyacrylamide gel electrophoresis (SDS-PAGE) and concentrated using an Amicon Ultra 30 kDa filter device (Merck Millipore, MA, USA).

### Purification of tRNA transcripts
Wild-type and mutant human tRNA^Phe transcripts were prepared using in vitro transcription. First, the DNA template was prepared using PCR with KOD-Plus-Neo polymerase (Toyobo, Osaka, Japan) using the primers listed in Supplementary Table 2. Thereafter, in vitro transcription was performed using T7 polymerase (New England Biolabs, MA, USA) and the prepared DNA template. Synthesized tRNA^Phe transcripts were purified via gel electrophoresis.

### Cryo-electron microscopy grid preparation
To obtain the cryo-EM structure of FTSJ1-THADA, the concentration of FTSJ1-THADA was adjusted to approximately 10 μM. To obtain the cryo-EM structure of FTSJ1-THADA-tRNA complex, purified FTSJ1-THADA (f.c 5 μM) was incubated with transcribed human cytoplasmic tRNA^Phe (f.c 10.4 μM) with or without S-adenosyl homocysteine (SAH) (f.c 1 mM) in reaction buffer (60 mM Tris-HCl (pH 7.5), 75 mM NaCl, 50 mM KCl, 5 mM $MgCl_2$, 1 mM DTT) at 37 °C for 30 min, and subsequently cooled on ice for approximately 30 min. Thereafter, 4 μl of each sample was applied to glow-discharged (5 mA for 30 s) Quantifoil R1.2/1.3 300 mesh Cu grids (Quantifoil, TH, Germany) and incubated at 4 °C for 3 s in a controlled environment of 100% humidity. Excess solution was blotted off with filter paper for 3 s and subsequently immediately plunged into liquid ethane for vitrification using Vitrobot Mark IV (Thermo Fisher Scientific, MA, USA).

### Cryo-electron microscopy data collection and processing
Micrographs were obtained using a Titan Krios transmission electron microscope (Thermo Fisher Scientific, MA, USA) operated at an accelerating voltage of 300 kV and equipped with a K3 camera (Gatan, CA, USA). Automated image acquisition was performed at the nominal magnification of ×105,000, corresponding to an objective pixel size of 0.8285 Å or 0.83 Å at the specimen level (Table 1). The total exposure of electrons at the specimen level was approximately 50 electrons/$Å^2$ and was fractionated into 48 frames (Table 1). Images were recorded as movie micrographs. Image processing was performed using RELION-3.1.3[59]. Details of image processing for each sample are presented in Supplementary Fig. 3. The movie micrographs obtained were motion corrected using the program implemented in RELION-3.1.3. Contrast transfer function (CTF) parameters were estimated using CTFFIND4.1[60]. Initial particle selection was performed using crYOLO-1.8.2[61], with a box size of 260 pixels. Obtained particles were binned to 150 pixels and subjected to 2D classification to discard junk particles. The remaining particles were subjected to 3D classification using the initial model generated in RELION-3.1.3. In processing FTSJ1-THADA data (Supplementary Fig. 3A), a consensus map that roughly fits the alpha fold prediction of THADA[28,29]. However, the map shows weak density for the N-terminal tail-like domains and the attached FTSJ1. Using multiple 3D and focused classifications targeting the FTSJ1 moiety at the original pixel size, the particles were separated into five classes with different arrangements of the N-terminal tail-like domains. In addition, Bayesian polishing, CTF refinement, and another round of auto-refinement and post-processing were performed. The final resolution achieved was 3.31–4.06 Å (Supplementary Fig. 4a). During FTSJ1-THADA data processing with a tRNA substrate (Supplementary Fig. 3b), particles were separated by the presence of a tRNA-like moiety in the first 3D classification. The tRNA-bound particles were subjected to 3D auto-refinement, followed by a second 3D classification to discard junk particles. The selected particles were subjected to 3D auto-refinement of their original pixel sizes. In addition, Bayesian polishing, CTF refinement, and another round of auto-refinement and post-processing were performed. The final resolution achieved was 2.66 Å (Supplementary Fig. 4b). The local resolution of each map was estimated using RELION-3.1.3 (Supplementary Fig. 4c–e). Moreover, the 3D continuous conformational heterogeneity was evaluated via a 3D variability analysis in cryoSPARC[62].

### Cryo-electron microscopy model building
The initial atomic model of FTSJ1(Q9UET6) or THADA (Q6YHU6) was obtained from the AlphaFold database[28,29]. The atomic model of tRNA^Phe was derived from a portion of the P-site tRNA of the *E. coli* 70S ribosome complex (PDB:7Y7F)[7]. The tRNA sequences were manually modified using the COOT-0.9.5[63]. The ligand restraint file for SAH was generated using eLBow[64], equipped with Phenix-1.18[65].

The atomic model of THADA was initially rigid-body fitted into the obtained map using UCSF Chimera v.1.14[66]. The remaining portion of the map was attributed to FTSJ1 (upper portion) or tRNA^Phe (internal portion). SAH was fitted to the remaining density near C32 of the tRNA using COOT-0.9.5[63]. This model was manually corrected using UCSF Chimera v.1.14[66] and COOT-0.9.5[63], followed by the final phenix.real space refinement routine using Phenix-1.18[65]. The final refined model was validated

using MOLPROBITY[67]. The map vs. model Fourier shell correlation (FSC) was calculated using Phenix-1.18[65] (Fig. S4F). Graphical figures were prepared using UCSF Chimera-X v.1.3[68].

## In vitro methylation assay

pcDNA3.4 vectors (Thermo Fisher Scientific, MA, USA) harboring the THADA mutant were constructed by quick-change site-directed mutagenesis using the primers listed in Supplementary Table 2. FLAG-tagged FTSJ1, THADA, or AHCY was overexpressed in Expi293 cells (Thermo Fisher Scientific, MA, USA) and purified using FLAG antibody-conjugated beads (product number: A2220, Merck Millipore, MA, USA) and poly FLAG peptides (product number: B23112, Selleck Chemicals, TX, USA) according to the manufacturer's instructions (https://www.sigmaaldrich.com/US/en/technical-documents/protocol/protein-biology/protein-purification/flag-immuno-precipitation-protocol). Total RNAs were extracted from FTSJ1 KO 293FT cells using the TRIzol reagent (Thermo Fisher Scientific, MA, USA) following the manufacturer's protocol. The FTSJ1 KO 293FT cell line was established by transient transfection of TrueGuide sgRNA (sgRNA ID = CRISPR896843_SGM, Thermo Fisher Scientific, MA, USA) and TrueCut Cas9 Protein v2 (Thermo Fisher Scientific, MA, USA) using the TransIT-X2 Dynamic Delivery System (Takara Bio, Shiga, Japan). Purified FTSJ1 (500 ng), THADA (3000 ng), and total RNAs (5000 ng) extracted from FTSJ1 KO cells were mixed in a reaction buffer (20 mM Tris-HCl (pH 7.4), 150 mM NaCl) and transferred to a 96-well plate for 30 min at room temperature (approximately 25 °C). A fluorescence developing buffer (final concentration: 20 μM SAM, 3 μM AHCY, 3 μM β-nicotinamide adenine dinucleotide (β-NAD) (Fujifilm Wako, Osaka, Japan), 0.5 U/μL adenosine deaminase (Fujifilm Wako, Osaka, Japan), 15 μM Thiol Fluorescent Probe IV (Sigma-Aldrich, MO, USA), 25 mM $KH_2PO_4$, 2 mM $MgCl_2$, 0.01% Triton X-100) was added to the complex. After 20 min, fluorescence intensity was measured using FlexStation3 (Molecular Devices, Tokyo, Japan) at an excitation wavelength of 400 nm and an emission wavelength of 465 nm. The fluorescence intensity obtained from the sample lacking AHCY/β-NAD was used as a baseline to normalize the intensities of the other samples.

## Immunoprecipitation

In brief, 293FT cells were transfected with plasmids expressing FLAG-tagged THADA using TransIT-LT1 reagent (Takara Bio, Shiga, Japan). After 3 days, the cells were lysed in a cell lysis buffer (20 mM Tris-HCl (pH 7.4), 150 mM NaCl, 1 mM EDTA, 1 mM EGTA, 1× cOmplete protease inhibitor, 0.01% Triton X-100). FLAG-tagged proteins were immunoprecipitated using an anti-FLAG M2 affinity gel (Sigma-Aldrich, MO, USA). The immunoprecipitants were boiled with SDS sample buffer (240 mM Tris-HCl (pH 6.8), 8% SDS, 40% glycerol, 0.1% bromophenol blue, 20% 2-mercaptoethanol) at 95 °C for 5 min. Equal volumes of samples were subjected to SDS-PAGE and blotted on a polyvinylidene difluoride membrane (Millipore, MA, USA). The membranes were blocked with 0.5% skim milk (Nacalai Tesque, Kyoto, Japan) in TBST buffer (137 mM NaCl, 2.68 mM KCl, 25 mM Tris (pH 7.4), 0.1% Tween20) at room temperature (approximately 25 °C) for 1 h and incubated with primary antibodies (anti-FLAG antibody (product number: 20543-1-AP, Proteintech, IL, USA) or anti-FTSJ1 antibody (product number: 11620-1-AP, Proteintech, IL, USA)) for overnight at 4 °C. After a brief wash with TBST three times, the membranes were incubated with horseradish peroxidase-conjugated secondary antibodies (product number: 7074, Cell Signaling Technology, MA, USA) for 1 h at room temperature. The signals were developed with Clarity Western enhanced chemiluminescence substrate (Bio-Rad Laboratories, CA, USA).

## Figure preparation

All figures were prepared using Canvas X (Nihon Poladigital, Tokyo, Japan). USCF Chimera X[68] was used to generate figures of cryo-EM maps and atomic models.

## Statistics and reproducibility

Replicates and statistical tests are described in the figure legends. No statistical methods were used to predetermine sample size. Data are presented as mean ± SD, as indicated in the figure legend. $P < 0.05$ was considered to be statistically significant.

## Reporting summary

Further information on research design is available in the Nature Portfolio Reporting Summary linked to this article.

## Data availability

Publicly available datasets from the Protein Data Bank (7Y7F and 6JPL) were used for atomic model building and comparison. Cryo-EM maps and atomic coordinates of reported structures were deposited in Electron Microscopy Data Bank (EMDB) and Protein Data Bank, respectively, with the following accession codes: EMD-38859 and 8Y2O (FTSJ1-THADA with tRNA); EMD-61701 (FTSJ1-THADA, Class 1); EMD-61702 (FTSJ1-THADA, Class 2a); EMD-61703 (FTSJ1-THADA, Class 2b); EMD-61704 (FTSJ1-THADA, Class 2c); EMD-61705 (FTSJ1-THADA, Class 2d).

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

## Acknowledgements

We thank K. Hanada and M. Yonemochi for technical support. All cryo-EM data were collected at the RIKEN Yokohama cryo-EM facility (Yokohama, Japan); thus, we also thank T. Uchikubo-Kamo and R. Akasaka for their assistance with the cryo-EM data collection and analysis. This study was supported by a grant-in-aid for scientific research from JSPS (20J00947 to K.I.), AMED (JP20cm0106179 to A.F.; JP22ama221001 to M.S.), and RIKEN (pioneering project 'Biology of Intracellular Environments' and BDR Structural Cell Biology Project to M.S.).

## Author contributions

K.I. prepared most materials and conducted cryo-EM analyses, which were supported by M.S.; A.F. conducted biochemical experiments. K.I. and M.S. wrote the manuscript. All authors discussed the results. M.S. designed and supervised the study.

## Competing interests

The authors declare no competing interests.
