## [Transparent Peer Review file · Communications Biology]

Structural insights into tRNA recognition of the human FTSJ1-THADA complex

Corresponding Author: Professor Mikako Shirouzu

Version 0:

Reviewer comments:

Reviewer #1

(Remarks to the Author)

General comments:

This manuscript presents structural analyses of the human FTSJ1-THADA complex, focusing on its interaction with tRNA. Using cryo-electron microscopy (cryo-EM), the authors elucidate how THADA anchors tRNA and identify residues essential for substrate recognition and Nm32 methylation. The study compares FTSJ1-THADA to the yeast FTSJ1-WDR6 complex, offering novel insights into substrate-specific interactions and highlighting structural adaptations.

1. Major Claims:

- 1) FTSJ1 binds THADA via a unique interaction mechanism distinct from WDR6.
- 2) The THADA component forms a cylindrical structure that recognizes and anchors tRNA substrates.
- 3) Residues critical for THADA-tRNA binding and enzymatic activity are identified.
- 4) A stepwise model of Nm32 and Nm34 modifications mediated by FTSJ1-THADA and FTSJ1-WDR6 is proposed.

Assessment: The claims are significant, novel, and contribute substantially to understanding tRNA modifications. The structural elucidation of the FTSJ1-THADA-tRNA complex is particularly compelling, filling a gap in the literature on tRNA methylation mechanisms in higher eukaryotes.

2. Novelty and Interest:

The manuscript offers new insights into the molecular basis of tRNA recognition by the FTSJ1-THADA complex. The findings will appeal to researchers in RNA biology, structural biology, and translational regulation. By integrating cryo-EM with biochemical and mutational studies, this work sets a new standard for exploring tRNA-protein interactions.

3. Strength of Evidence:

• Convincing evidence:

- 1) High-resolution cryo-EM maps and models (2.66 Å) validate structural claims.
- 2) Mutation studies robustly link specific residues to tRNA binding and enzymatic activity.
- 3) Comparative structural analysis with FTSJ1-WDR6 strengthens the conclusions about interaction differences.

• Limitations:

- 1) The functional implications of Nm32 modification on cellular or organismal phenotypes remain speculative. Follow-up studies might be needed to fully delineate its physiological role.
- 2) The modeling of flexible regions (e.g., THADA's N-terminal tail-like domain) is incomplete, potentially impacting conclusions on open/closed conformations.

4. Statistical Analyses and Reproducibility:

- 1) Statistical tests (e.g., ANOVA) are appropriate for mutational activity assays, but variability (e.g., error bars in Fig. 5f) warrants further replication.
- 2) The methodology is meticulously detailed, supporting reproducibility. Still, providing raw cryo-EM data in accessible repositories would enhance transparency.

5. Broader Impact:

This study significantly advances the understanding of RNA modifications. By dissecting how THADA supports Nm32 modification, it broadens our grasp of FTSJ1-associated disorders (e.g., intellectual disability). The structural insights might

inform drug development targeting RNA methylation pathways.

Specific Comments and Suggestions for Improvement:

1. Introduction:

o The introduction effectively sets the stage but could benefit from a clearer delineation of the biological significance of Nm32 and Nm34 modifications in humans.

2. Results:

1) Structural claims on open vs. closed forms: The rationale for assigning certain states as "open" or "closed" requires further clarification. Are these states physiologically relevant, or artifacts of cryo-EM preparation?

2) Mutation data: Expand on why some mutations (e.g., Glu1225Ala) show modest effects. Could redundancy in the binding network explain this?

3. Figures:

1) Figures are well-prepared but often too dense. Simplified schematic diagrams summarizing tRNA binding mechanisms and structural changes in FTSJ1-THADA would aid comprehension.

2) The electrostatic maps (Fig. 3f, 3g) are informative but require clearer labeling to highlight key functional regions.

4. Discussion:

o While comprehensive, the discussion lacks a clear roadmap for future research. For example:

1) How could these findings inform therapeutic strategies for intellectual disabilities linked to FTSJ1 mutations?

2) What experimental approaches could validate the proposed stepwise model for Nm32 and Nm34 modifications?

Reviewer #2

(Remarks to the Author)

Ishiguro and colleagues present a structural study of the methyltransferase FTSJ1 and an accessory factor THADA, that allows the methyltransferase to specifically target methylation to bases at position 32 in tRNAs. The work complements prior studies on a FTSJ1 ortholog in complex with a different accessory factors, WDR6, which directs FTSJ1 to base 34.

The structural study is highly informative about conformational changes that occur when the complex shifts from a free enzyme to binding its tRNA substrate. The large surface area of THADA complements the shape of the tRNA and rearranges the conformation of the anticodon loop to place base 32 in the active site of the methyltransferase. Overall, this study provides new insights into tRNA recognition and how this is used to direct a highly specific methylation event that is important for cell growth.

The presentation of the structural data is good and the data and model for the complex seem to be of good quality. The structures of the free enzyme are somewhat lower resolution as a consequence of structural heterogeneity and the authors do not over-interpret the information available from these datasets.

The structural work is followed up by an analysis of THADA mutants with a methyltransferase assay and this is important to show that altered residues important for tRNA recognition affect activity. However, this part of the work is somewhat weak, in that there is no additional follow up biochemistry to show that the mutant THADA proteins are correctly folded. For example, showing that these mutants still bind to FTSJ1 by size exclusion chromatography or a pull-down assay. Also, regarding the assay, the authors state that "... mutations in Arg199, ... and Tyr843 almost completely abolished methyltransferase activity." While methyltransferase activity is certainly lower in these mutants ($p < 0.05$), there are still varying levels of fluorescence recorded and it is not clear how this would differ from background fluorescence e.g. if the assay was done with THADA but no FTSJ1. Ideally, the authors should provide this background data alongside that of the mutants and/or modify their statement.

While the structural work is carried out with a tRNA^{Phe}, the authors note that this is not the only tRNA to be methylated by FTSJ1. In the discussion they present an analysis of previously published data to show that other tRNAs have sequence biases at specific positions and that this fits with the current structural analysis that of bases that are important for recognition and placement of base 32 in the active site. It's not clear to me why this analysis was placed in the discussion, rather than the main text. Ideally, it would be accompanied by a structural figure to indicate the positions of the key bases that show strong biases. It would also be great to see one or two other tRNAs modelled into tRNA binding site to give an indication of how these biases would influence/support binding.

On a minor point, Figure 4 is difficult to navigate. The authors should indicate on the figure the relationship between sites in Figure 4a and the zoomed in Figures 4c-e.

Reviewer #3

(Remarks to the Author)

Brief Summary of Manuscript: In the manuscript (submitted for consideration as an Article in the journal Communications Biology) "Structural insights into tRNA recognition of the human FTSJ1-THADA complex" by Ishiguro et. al, the authors solve the structures of the human FTSJ1-THADA and the FTSJ1-THADA-tRNA^{Phe} complexes using Cryo-EM. Based on the structure, they then generate several THADA variants and test their methyltransferase activity. This manuscript presents the first structural data on THADA (Trm732 in yeast), the FTSJ1-THADA complex, and the first of either FTSJ1 complex bound to tRNA.

Overall Impression of the work: This report represents a significant step forward in our understanding of how Nm32 and Nm34 are formed in eukaryotes, and for tRNA modification in general. This manuscript presents the first structural data on THADA (Tm732 in yeast), the FTSJ1-THADA complex, and the first of either of the FTSJ1 complexes bound to tRNA. In my opinion, the manuscript meets all of the criteria for the aim and scope of the journal, including novelty, strong evidence for its conclusions, strong data, and importance to the protein translation and tRNA modification sub-field of biology. The Cryo-EM data seems strong and well-done. However, there were some minor issues with the methyltransferase assays. This part of the manuscript should be clarified to state exactly what is being measured in the assays. See below for comments on this topic and others that could improve and strengthen this important manuscript.

Specific Comments:

Abstract and Intro are overall well-written. A few minor suggestions:

1. In the abstract, there is a missing "are" between lines 23 and 24
2. On line 43, the sentence starts with "Recently" However, most of the papers cited are not very recent, with one of them from 2004

Results does a good job of explaining the results.

3. Lines 95 – 97 state that size-exclusion chromatography purification of the complex was performed and shown in figure 1b. What is the size of the complex as analyzed here? Is it a simple heterodimer (~250 kDa), or is there evidence of larger complexes? Please comment in the manuscript or add gel filtration size markers to SDS-PAGE.
4. Line 190 states "formed hydrogen bonds with the base of the tRNA". I believe it should state that the residues formed H-bonds with the bases (plural) (unless authors are referring to something termed the "base of the tRNA")
5. Lines 198-205I believe that figures 5e and 5f are mislabeled. 5f is the in vitro activity and 5e is the figure with motifs 1 and 2 of THADA
6. Some further clarification is needed in the parts described in line 197 – 198, which corresponds to fig 5f and the methyltransferase activity assays. It is not clear exactly how the activity assay works as described. In the results section, it should be noted that the assay was performed on total tRNA with a fluorescent substrate, and exactly what is measured by the fluorescent substrate. In the methods section, please describe how total tRNA was purified, as well as how the assay works. I assume that it is directly measuring the formation of SAH, which is produced when the SAM is demethylated by the FTSJ1-THADA complex upon transferring a methyl group to another molecule (almost certainly a tRNA substrate) in the total tRNA prep. It should be clear that it is an indirect measure of tRNA methylation activity and that methylation of specific tRNAs is not actually being measured.

Discussion is well-written. I have a few suggestions to help improve this section:

Throughout the discussion, there are several instances where references are missing and/or could help clarify what is being stated. Also, when describing results from the paper, it would really help the reader to refer back to the figure that shows the result. Below are some instances of missing references or figures

7. Line 223, clause ending in tRNAPhe should have a reference
8. In lines 233 and 234, it is not clear if this is something noted by the authors of the current manuscript or something pointed out in reference 21. Please clarify
9. Line 230 – not sure that THADA and WDR6 should be classified as enzymes because they do not have enzymatic activity. Adaptor proteins might be a better way to describe them.
10. Lines 234 -236. Was the Cys238 observation first pointed out in the current manuscript or in reference 21? I assume so, but it is not clear as written. Perhaps putting the statement in present tense will help clarify. Also, please remind the reader why Cys238 is of interest based on the Cryo-EM structure and give the figure reference so the reader can refer back.
11. Change sentence that starts on line 237 and ends on 238 from describing the tRNA as "FTSJ1-THADA substrates" to something such as "tRNAs containing Nm32 modifications". Necessity of THADA for these tRNAs has not been formally shown.
12. Lines 241 – 245. As written, the argument is not entirely clear. Need to explicitly remind the reader that Tyr843 and Thr798 interact with A36 and A38 and refer back to figure 5C.
13. Lines 250 – 262 need to reference supplementary figure 8.
14. Line 288. Do you mean THADA having a function in fertility? PCOS effects many other aspects of fertility than just specifically pregnancy.

Version 1:

Reviewer comments:

Reviewer #1

(Remarks to the Author)

The authors have addressed most of my concerns; the manuscript has been significantly improved.

Reviewer #2

(Remarks to the Author)

The changes made to the manuscript have addressed my comments and substantively increased the quality of the data presented and their interpretation. I am happy to recommend the work for publication.

Reviewer #3

(Remarks to the Author)

In the manuscript "Structural insights into tRNA recognition of the human FTSJ1-THADA complex" by Ishiguro et. al, the authors solve the structures of the human FTSJ1-THADA and the FTSJ1-THADA-tRNAPhe complexes using Cryo-EM. Based on the structure, they then generate several THADA variants and test their methyltransferase activity. This manuscript presents the first structural data on THADA (Trm732 in yeast), the FTSJ1-THADA complex, and the first of either FTSJ1 complex bound to tRNA. This report represents a significant step forward in our understanding of how Nm32 and Nm34 are formed in eukaryotes, and for tRNA modification in general. This manuscript presents the first structural data on THADA (Trm732 in yeast), the FTSJ1-THADA complex, and the first of either of the FTSJ1 complexes bound to tRNA. In my opinion, the manuscript meets all of the criteria for the aim and scope of the journal, including novelty, strong evidence for its conclusions, strong data, and importance to the protein translation and tRNA modification sub-field of biology. The Cryo-EM data seems strong and well-done. In the second round of submission, the authors adequately addressed my concerns as a reviewer, and I feel that they also adequately addressed the concerns of the other reviewers. When published, this work will significantly advance our understanding of how FTSJ1 works with its protein partners to modify the anticodon loop of tRNA.

To whom it may concern,

Thank you for the opportunity to revise our manuscript titled “Structural insights into tRNA recognition of the human FTSJ1-THADA complex” (COMMSBIO-24-7654-T). In the revised manuscript, we added experimental data as suggested and carefully revised the manuscript following the reviewers’ comments. Below are our point-by-point responses to each of the comments. Text in blue is the original comments from the reviewers and our responses are in black.

During the revision, we noticed that the Cryo-EM model is not rightly superimposed on maps in several figures (Fig. 2ef and S5). The N-terminal tail-like domain in class 2a–2d further shifted outward, as much as 24–52 Å, measured from helix $\alpha 1$ at the end of the N-terminal tail-like domain (Page 5 lines 23–25).

Fig.2

Fig.S5

Reviewer #1 (Remarks to the Author):

General comments:

This manuscript presents structural analyses of the human FTSJ1-THADA complex, focusing on its interaction with tRNA. Using cryo-electron microscopy (cryo-EM), the authors elucidate how THADA anchors tRNA and identify residues essential for substrate recognition and Nm32 methylation. The study compares FTSJ1-THADA to the yeast FTSJ1-WDR6 complex, offering novel insights into substrate-specific interactions and highlighting structural adaptations.

1. Major Claims:

- 1) FTSJ1 binds THADA via a unique interaction mechanism distinct from WDR6.*
 - 2) The THADA component forms a cylindrical structure that recognizes and anchors tRNA substrates.*
 - 3) Residues critical for THADA-tRNA binding and enzymatic activity are identified.*
 - 4) A stepwise model of Nm32 and Nm34 modifications mediated by FTSJ1-THADA and FTSJ1-WDR6 is proposed.*
- Assessment: The claims are significant, novel, and contribute substantially to understanding tRNA modifications. The structural elucidation of the FTSJ1-THADA-tRNA complex is particularly compelling, filling a gap in the literature on tRNA methylation mechanisms in higher eukaryotes.*

2. Novelty and Interest:

The manuscript offers new insights into the molecular basis of tRNA recognition by the FTSJ1-THADA complex. The findings will appeal to researchers in RNA biology, structural biology, and translational regulation. By integrating cryo-EM with biochemical and mutational studies, this work sets a new standard for exploring tRNA-protein interactions.

Re: Thank you for your thoughtful and positive review of our manuscript. We appreciate your recognition of the significance and novelty of our findings regarding the FTSJ1-THADA complex. Your comments on the structural elucidation of the complex and its implications for tRNA methylation mechanisms are highly encouraging. Your feedback reinforces the importance of our work, and we addressed each of your suggestions in the revised manuscript to enhance our work.

3. Strength of Evidence:

• Convincing evidence:

- 1) High-resolution cryo-EM maps and models (2.66 Å) validate structural claims.*
- 2) Mutation studies robustly link specific residues to tRNA binding and enzymatic activity.*
- 3) Comparative structural analysis with FTSJ1-WDR6 strengthens the conclusions about interaction differences.*

• Limitations:

- 1) The functional implications of Nm32 modification on cellular or organismal phenotypes remain speculative. Follow-up studies might be needed to fully delineate its physiological role.*
- 2) The modeling of flexible regions (e.g., THADA's N-terminal tail-like domain) is incomplete, potentially impacting*

conclusions on open/closed conformations.

Re: We agree with these limitations. We added some experiments necessary to clarify these points in the discussion in the future.

- 1) THADA is associated with various cellular processes and diseases, but whether these functions are derived from tRNA modification activity or other functions of THADA (Page 11 line 3–Page 12 line 5) is unclear. Our structural analysis reveals important residues contributing directly to tRNA modification activity, enabling us to show whether these phenotypes are directly related to tRNA-modifying activity. For example, from the structural analysis, the SNP associated with T2DM may not be directly involved in tRNA-modifying activity (Page 11 lines 24–26), suggesting that T2DM is not directly related to Nm32 modification. Further physiological and biochemical analyses based on our structural analyses can help elucidate the roles of FTSJ1-THADA and Nm32 in these diseases (Page 11 line 35–Page 12 line 5).
- 2) Because the structures of FTSJ1-THADA without tRNA substrate did not achieve atomic resolution for the N-terminal tail-like domain, atomic models of these structures could not be made. To evaluate the conformational heterogeneity of these structures more clearly, we performed a 3D variability analysis using all THADA-like particles (Movie S1), Class 1 particles (Movie S2), and Class 2 particles (Movie S3) from the dataset of FTSJ1-THADA without the tRNA substrate. The N-terminal tail-like domain of THADA was found to be highly flexible, showing movement between two different conformations, corresponding to the open or closed form (Page 5, lines 27–31). In contrast, the tRNA-bound FTSJ1-THADA was observed exclusively in the closed form (Movies S4) (Page 6, lines 21–23). We added a figure showing that the N-terminal tail-like domain of THADA could not contact the D-loop of tRNA in the open form, as indicated by superimposed model (Fig. S6) (Page 6, lines 23–26). These analyses showed the existence of two conformations of FTSJ1-THADA (open and closed forms) in the absence of tRNA. After the tRNA substrate was anchored, the N-terminal tail-like domain of THADA was stabilized in the closed form by the binding of the tRNA substrate.

Fig.S6

4. Statistical Analyses and Reproducibility:

1) Statistical tests (e.g., ANOVA) are appropriate for mutational activity assays, but variability (e.g., error bars in Fig. 5f) warrants further replication.

Re: We agree with this point. We performed the same experiment, including conditions without the FTSJ1 enzyme and without THADA, confirming data reproducibility with our original submission (Fig. 5e).

Fig.5

2) *The methodology is meticulously detailed, supporting reproducibility. Still, providing raw cryo-EM data in accessible repositories would enhance transparency.*

Re: We agree with this point. We will deposit raw cryo-EM data in EMPIAR after publication.

5. Broader Impact:

This study significantly advances the understanding of RNA modifications. By dissecting how THADA supports Nm32 modification, it broadens our grasp of FTSJ1-associated disorders (e.g., intellectual disability). The structural insights might inform drug development targeting RNA methylation pathways.

Re: Thank you for highlighting the broader impact of our study. In response to your valuable feedback, we expanded the discussion section to elaborate on the relevance of our structural insights to disease mechanisms and potential therapeutic applications (Page 10 line 30–Page 11 line 2 and Page 11 line 35–Page 12 line 5).

Specific Comments and Suggestions for Improvement:

1. Introduction:

o The introduction effectively sets the stage but could benefit from a clearer delineation of the biological significance of Nm32 and Nm34 modifications in humans.

Re: We added a sentence emphasizing the physiological significance of Nm32 and Nm34 (Page 3 lines 23–24).

2. Results:

1) *Structural claims on open vs. closed forms: The rationale for assigning certain states as "open" or "closed" requires further clarification. Are these states physiologically relevant, or artifacts of cryo-EM preparation?*

Re: We think the conformational change of THADA between “open” and “closed” forms has physiological significance. Both conformations were detected in the cryo-EM sample without tRNA substrate, but in the cryo-EM sample with tRNA substrate, the tRNA-bound class was only in the closed form. This suggests that both conformations are allowed without tRNA binding, but after tRNA binding, THADA is stabilized in closed form. Furthermore, stabilization to closed form contributes to the recognition of the tRNA substrate and enhanced

methyltransferase activity because the loss of interaction between THADA and tRNA via the N-terminal tail-like domain (Arg199Ala mutation) almost abolishes the methyltransferase activity of FTSJ1-THADA. We clarified these points in the discussion section (Page 9 lines 9–15).

2) Mutation data: Expand on why some mutations (e.g., Glu1225Ala) show modest effects. Could redundancy in the binding network explain this?

Re: Around the D-loop, Lys92 interacts with the backbone of U16 and may stabilize the tRNA position redundantly. The modest effect of Glu1225Ala mutation suggests that the Glu1225-G20 interaction is not a critical determinant for tRNA binding, but rather a facilitator. We added these points to the discussion (Page 10 lines 11–14).

To comprehensively understand the details of tRNA recognition by FTSJ1-THADA, kinetic analysis using many mutations in THADA (and also in tRNA) is needed. We added this point to the future perspective in the discussion section (Page 9 lines 14–16).

3. Figures:

1) Figures are well-prepared but often too dense. Simplified schematic diagrams summarizing tRNA binding mechanisms and structural changes in FTSJ1-THADA would aid comprehension.

Re: We added the information on key residues for tRNA binding in Fig. 6. The information about structural changes of FTSJ1-THADA during tRNA recognition (open and closed forms) is also included.

Fig.6

2) The electrostatic maps (Fig. 3f, 3g) are informative but require clearer labeling to highlight key functional regions.

Re: We added the position of the methyltransferase center, THADA binding site of FTSJ1, SAH, and key tRNA residues for tRNA-FTSJ1-THADA formation in Figs. 3f and 3g.

Fig.3

4. Discussion:

o While comprehensive, the discussion lacks a clear roadmap for future research. For example:

1) How could these findings inform therapeutic strategies for intellectual disabilities linked to FTSJ1 mutations?

Re: Although this paper does not directly contribute to therapeutic strategies, it may contribute to elucidating the physiological mechanism of intellectual disability associated with FTSJ1 mutation. For example, a novel missense allele of FTSJ1 (A26P) associated with intellectual disability reduces the frequency of Nm32 but not Nm34. A26 is located close to the tRNA substrate and SAM, but the SAM recognition mechanism is conserved between FTSJ1-THADA and FTSJ1-WDR6. Therefore, the difference in the tRNA recognition mechanism between FTSJ1-THADA and FTSJ1-WDR6 may explain why A26P only reduces the frequency of Nm32. Elucidation of the FTSJ1-WDR6-tRNA structure will lead to its understanding. This has been added to the discussion (Page 10 line 30–Page 11 line 2).

2) What experimental approaches could validate the proposed stepwise model for Nm32 and Nm34 modifications?

Re: In the second paragraph of the discussion, we added two experiments necessary in the future (Page 9 lines 20–23). The first approach is to examine the tRNA modification status in tRNA precursors. If our proposed model is correct, Nm32 would be found in both early and late precursors, but Nm34 would only be found in later precursors. The second approach is to examine the effect of Nm34 on tRNA recognition by FTSJ1-THADA. If the presence of Nm34 inhibits or significantly reduces the tRNA recognition by FTSJ1-THADA, it would be important evidence in support of our model.

We mentioned other future experiments in Discussion (Page 9 lines 13–15, Page 12 lines 3–5).

Reviewer #2 (Remarks to the Author):

Ishiguro and colleagues present a structural study of the methyltransferase FTSJ1 and an accessory factor THADA, that allows the methyltransferase to specifically target methylation to bases at position 32 in tRNAs. The work complements prior studies on a FTSJ1 ortholog in complex with a different accessory factors, WDR6, which directs FTSJ1 to base 34.

The structural study is highly informative about conformational changes that occur when the complex shifts from a free enzyme to binding its tRNA substrate. The large surface area of THADA complements the shape of the tRNA and rearranges the conformation of the anticodon loop to place base 32 in the active site of the methyltransferase. Overall, this study provides new insights into tRNA recognition and how this is used to direct a highly specific methylation event that is important for cell growth.

The presentation of the structural data is good and the data and model for the complex seem to be of good quality. The structures of the free enzyme are somewhat lower resolution as a consequence of structural heterogeneity and the authors do not over-interpret the information available from these datasets.

Re: Thank you for your detailed and constructive review. We appreciate your positive assessment of our structural analysis and its contribution to understanding tRNA recognition and methylation specificity. Your comments on the structural quality and conformational insights are highly encouraging.

The structural work is followed up by an analysis of THADA mutants with a methyltransferase assay and this is important to show that altered residues important for tRNA recognition affect activity. However, this part of the work is somewhat weak, in that there is no additional follow up biochemistry to show that the mutant THADA proteins are correctly folded. For example, showing that these mutants still bind to FTSJ1 by size exclusion chromatography or a pull-down assay. Also, regarding the assay, the authors state that "... mutations in Arg199, ... and Tyr843 almost completely abolished methyltransferase activity." While methyltransferase activity is certainly lower in these mutants ($p < 0.05$), there are still varying levels of fluorescence recorded and it is not clear how this would differ from background fluorescence e.g. if the assay was done with THADA but no FTSJ1. Ideally, the authors should provide this background data alongside that of the mutants and/or modify their statement.

Re: We acknowledge your concerns regarding the methyltransferase assay of THADA mutants. We performed immunoprecipitation experiments with FTSJ1 and wild-type or mutant THADA and demonstrated that they showed biochemical binding (Fig. S8). For the *in vitro* assay with the recombinant protein, we followed your suggestion and repeated the same experiments, including FTSJ1-free condition or THADA-free condition (Fig. 5e). We believe that these data indicate that our *in vitro* assay is valid and our interpretation is reasonable.

Fig.S8

Fig.5

While the structural work is carried out with a tRNA^{Phe}, the authors note that this is not the only tRNA to be

methyated by FTSJ1. In the discussion they present an analysis of previously published data to show that other tRNAs have sequence biases at specific positions and that this fits with the current structural analysis that of bases that are important for recognition and placement of base 32 in the active site. It's not clear to me why this analysis was placed in the discussion, rather than the main text. Ideally, it would be accompanied by a structural figure to indicate the positions of the key bases that show strong biases. It would also be great to see one or two other tRNAs modelled into tRNA binding site to give an indication of how these biases would influence/support binding.

Re: We appreciate your recognition of the importance of these data. However, we placed this analysis in the discussion because we consider these to be weak and the data cover only some tRNAs, not all tRNAs, and therefore may overlook Nm32-containing tRNAs. Consequently, we may misinterpret the tRNA sequence preferences of FTSJ1-THADA. A comprehensive analysis of human tRNA modifications has not been performed to date, and this is a limitation of the current research.

To clarify the relationship between this and the structural analysis, we mentioned structural data and Figures in this paragraph (Page 10 lines 4 and 7).

On a minor point, Figure 4 is difficult to navigate. The authors should indicate on the figure the relationship between sites in Figure 4a and the zoomed in Figures 4c-e.

Re: We amended Fig. 4a by adding a border to indicate the area shown in Fig. 4c-e.

We added helix numbers in Fig. 4c-e.

Fig.4

Reviewer #3 (Remarks to the Author):

Brief Summary of Manuscript: In the manuscript (submitted for consideration as an Article in the journal Communications Biology) “Structural insights into tRNA recognition of the human FTSJ1-THADA complex” by Ishiguro et. al, the authors solve the structures of the human FTSJ1-THADA and the FTSJ1-THADA-tRNAPhe complexes using Cryo-EM. Based on the structure, they then generate several THADA variants and test their

methyltransferase activity. This manuscript presents the first structural data on THADA (Trm732 in yeast), the FTSJ1-THADA complex, and the first of either FTSJ1 complex bound to tRNA.

Overall Impression of the work: This report represents a significant step forward in our understanding of how Nm32 and Nm34 are formed in eukaryotes, and for tRNA modification in general. This manuscript presents the first structural data on THADA (Trm732 in yeast), the FTSJ1-THADA complex, and the first of either of the FTSJ1 complexes bound to tRNA. In my opinion, the manuscript meets all of the criteria for the aim and scope of the journal, including novelty, strong evidence for its conclusions, strong data, and importance to the protein translation and tRNA modification sub-field of biology. The Cryo-EM data seems strong and well-done. However, there were some minor issues with the methyltransferase assays. This part of the manuscript should be clarified to state exactly what is being measured in the assays. See below for comments on this topic and others that could improve and strengthen this important manuscript.

Re: Thank you for your thoughtful and encouraging review of our manuscript. We are pleased that you recognize the novelty and significance of our work in advancing the understanding of tRNA modification and methylation. Your positive assessment of our structural data and their relevance to the field is highly appreciated.

We acknowledge your concerns regarding the methyltransferase assays and clarified our description. We carefully addressed your specific comments which has helped strengthen the manuscript. Your feedback is invaluable, and we appreciate your time and effort in reviewing our work.

Specific Comments:

Abstract and Intro are overall well-written. A few minor suggestions:

1. In the abstract, there is a missing “are” between lines 23 and 24

Re: We corrected it (Page 2 line 5).

2. On line 43, the sentence starts with “Recently” However, most of the papers cited are not very recent, with one of them from 2004

Re: We revised the sentence accordingly (Page 3 line 9).

Results does a good job of explaining the results.

3. Lines 95 – 97 state that size-exclusion chromatography purification of the complex was performed and shown in figure 1b. What is the size of the complex as analyzed here? Is it a simple heterodimer (~250 kDa), or is there evidence of larger complexes? Please comment in the manuscript or add gel filtration size markers to SDS-PAGE.

Re: We clarified the formation of the simple $\alpha\beta$ heterodimer (~260kDa) in the manuscript (Page 5 lines 4–7).

4. Line 190 states “formed hydrogen bonds with the base of the tRNA”. I believe it should state that the residues formed H-bonds with the bases (plural) (unless authors are referring to something termed the “base of the tRNA”)

Re: We corrected it (Page 8 line 4).

5. Lines 198-205I believe that figures 5e and 5f are mislabeled. 5f is the in vitro activity and 5e is the figure with motifs 1 and 2 of THADA

Re: We corrected the figure numbers.

Fig.5

6. Some further clarification is needed in the parts described in line 197 – 198, which corresponds to fig 5f and the methyltransferase activity assays. It is not clear exactly how the activity assay works as described. In the results section, it should be noted that the assay was performed on total tRNA with a fluorescent substrate, and exactly what is measured by the fluorescent substrate. In the methods section, please describe how total tRNA was purified, as well as how the assay works. I assume that it is directly measuring the formation of SAH, which is produced when the SAM is demethylated by the FTSJ1-THADA complex upon transferring a methyl group to another molecule (almost certainly a tRNA substrate) in the total tRNA prep. It should be clear that it is an indirect measure of tRNA methylation activity and that methylation of specific tRNAs is not actually being measured.

Re: We appreciate your valuable comments. We followed your suggestion and added a description of the mechanism of the in vitro methyltransferase assay in the main text: “In this assay, SAH produced by the methyltransferase activity of FTSJ1 was degraded to adenosine and homocysteine by adenosylhomocysteinase (AHCY). By detecting the amount of homocysteine with the fluorescence intensity of the thiol fluorescent probe IV, the methyltransferase activity of FTSJ1-THADA was semi-quantitatively quantified.” (Page 8 lines 12–16) We added the purification procedure of total RNA (Page 15 line 33–Page 16 line 3). We used total RNA instead of specific tRNAs as the substrate for this assay. Although the use of total RNA as a substrate may increase the possibility of observing non-specific reactions, our setup was adequate to measure the enzymatic activity of FTSJ1 and depict the effect of the point mutation in THADA, judged by the measurements in the THADA-free condition and in the FTSJ1-free condition added as the negative control (Fig. 5e). However, as you pointed out, we understand that this setup does not measure methylation of specific tRNAs, which is a limitation.

Fig.5

Discussion is well-written. I have a few suggestions to help improve this section:

Throughout the discussion, there are several instances where references are missing and/or could help clarify what is being stated. Also, when describing results from the paper, it would really help the reader to refer back to the figure that shows the result. Below are some instances of missing references or figures

Re: We agree with this point. We added the relevant references and figure references in the discussion section.

7. Line 223, clause ending in tRNAPhe should have a reference

Re: We added the relevant references (Page 9 line 18).

8. In lines 233 and 234, it is not clear if this is something noted by the authors of the current manuscript or something pointed out in reference 21. Please clarify

Re: We clarified this point. While the sequence alignment between FTSJ1_Nm32 and FTSJ1_Nm34 was already performed in a previous study (reference 21), the observation regarding Cys238 is described for the first time in this study. We added the reference of Fig. 4, showing FTSJ1 residues participating in the interaction with THADA (Page 9 lines 28–31).

9. Line 230 – not sure that THADA and WDR6 should be classified as enzymes because they do not have enzymatic activity. Adaptor proteins might be a better way to describe them.

Re: We corrected it (Page 9 line 26).

10. Lines 234 -236. Was the Cys238 observation first pointed out in the current manuscript or in reference 21? I assume so, but it is not clear as written. Perhaps putting the statement in present tense will help clarify. Also, please remind the reader why Cys238 is of interest based on the Cryo-EM structure and give the figure reference so the reader can refer back.

Re: We have clarified this point (Page 9, lines 31–32).

11. Change sentence that starts on line 237 and ends on 238 from describing the tRNA as “FTSJ1-THADA substrates” to something such as “tRNAs containing Nm32 modifications”. Necessity of THADA for these tRNAs has not been formally shown.

Re: We agree with this point. We have corrected it (Page 9 line 35).

12. Lines 241 – 245. As written, the argument is not entirely clear. Need to explicitly remind the reader that Tyr843 and Thr798 interact with A36 and A38 and refer back to figure 5C.

Re: We mentioned the interaction with Tyr843 and Thr798 along with the relevant figure references (Page 10 line 4).

13. Lines 250 – 262 need to reference supplementary figure 8.

Re: We added the relevant figure references in this paragraph (Page 10 lines 16, 20, 24–26, and 29). The figure is renumbered to supplementary figure 10.

14. Line 288. Do you mean THADA having a function in fertility? PCOS effects many other aspects of fertility than just specifically pregnancy.

Re: We agree with this point. The evidence for a relationship between THADA and PCOS is not strong due to conflicting results. So we toned down the wording in this section (Page 11 lines 27–29).